# Optical Properties of Electrically Active Gold Nanoisland Films Enabled with Interfaced Liquid Crystals

**DOI:** 10.3390/nano10020290

**Published:** 2020-02-09

**Authors:** Hung-Chi Yen, Tsung-Rong Kuo, Chun-Ta Wang, Jia-De Lin, Chia-Chun Chen, Yu-Cheng Hsiao

**Affiliations:** 1Department of Chemistry, National Taiwan Normal University, 88 Ting-Chow Rd., Sec. 4, Taipei 11677, Taiwan; hungchiyen@gmail.com (H.-C.Y.); cjchen@ntnu.edu.tw (C.-C.C.); 2Graduate Institute of Nanomedicine and Medical Engineering, Taipei Medical University, 250 Wuxing St., Taipei 11031, Taiwan; trkuo@tmu.edu.tw; 3International PhD Program for Biomedical Engineering, Taipei Medical University, 250 Wuxing St., Taipei 11031, Taiwan; 4Department of Photonics, National Sun Yat-Sen University, 70 Lienhai Rd., Kaohsiung 80424, Taiwan; wangchunta4@gmail.com; 5Department of Engineering Science, University of Oxford, Parks Road, Oxford OX1 3PJ, UK; geman1218@yahoo.com.tw; 6Graduate Institute of Biomedical Optomechatronics, Taipei Medical University, 250 Wuxing St., Taipei 11031, Taiwan

**Keywords:** liquid crystal, optical device, gold nanoisland film

## Abstract

A system comprising a gold nanoisland film (Au NIF) covered with a liquid crystal (LC) material is introduced. By applying a voltage across the LC bulk, we demonstrate that changes in the refractive-index and orientation significantly modified the hybrid plasmonic–photonic resonances of the Au NIF. The hybrid structure enabled active control of the spectrum of the resonance wavelength of the metallic nanoisland by means of an externally applied electric field. Our modeling supports the observed results in LC/Au NIF. In a combination of the nanostructured surface with birefringent LCs, nonpolarized wavelength tunability of ~15 nm and absorbance tunability of ~0.024 were achieved in the visible wavelength, opening the door to optical devices and nanoscale sensors.

## 1. Introduction

Gold nanoisland films (Au NIFs) with nanosized gaps have been fabricated for applications in imaging and sensing due to their unique surface plasmon resonances [1,2,3]. Au NIFs with surface plasmon resonances in the near-infrared (NIR) region have been utilized to enhance the fluorescence intensity of various NIR fluorophores for imaging and sensing applications. For example, biochips of Au NIFs captured immunoglobulin G (IgG) and IgA antibodies against Zika virus or dengue virus antigens in human serum and then were conjugated with fluorophores of anti-human IgG-IRDye680 and anti-human IgA-IRDye800 [4]. An Au NIF platform showed the ability to amplify the NIR fluorescence of fluorophores of anti-human IgG-IRDye680 and anti-human IgA-IRDye800 by about 100-fold, allowing for highly sensitive detection of Zika virus infection and dengue virus infection. Although Au NIFs have been extensively applied for imaging and sensing applications, the development of Au NIFs of a suitable wavelength and intensity of surface plasmon resonance is still being processed to optimize detection conditions. It is practically significant to enable active control of these plasmon resonance properties by external tuning methods. Such tunability can be achieved by applying voltage, heat, or an illumination profile after incorporating other materials [5,6,7]. Recently, the electrical tuning of nano-antenna resonances in the visible regime was demonstrated [8].

Liquid crystals (LCs) are interesting materials because their electrically induced reorientation of molecules can modify the resonance conditions of optical resonators. Indeed, LCs can be employed to control resonances of metallic nanostructures, including localized surface plasmon resonance (LSPR)-based nano-antennas and surface plasmon polaritons in metallic films [9,10,11,12]. In addition, many plasmonic response combined with LC devices have been proposed in the past [13,14,15]. To the best of our knowledge, a hybrid LC/nanoisland structure has not been reported until now. The tuning ranges of LC-based metallic nanostructures are quite limited in polarized light. However, polarization-independent LC-based plasmonic surfaces have been invented with a large range of the colour spectrum [16]. In addition, the large tuning range of hybrid LC/metallic nanostructures are also proposed in the literature [17].

In this study, we first show the active tuning of Au NIF resonances in a metallic nanostructure embedded in an LC matrix. Tuning was achieved by applying an electric field to the LC/Au NIF device confined by an Au nanostructured substrate and a conducting glass substrate coated with polyimide (PI) for planar alignment of the LC molecules. As evidenced by the voltage-dependent nonpolarized transmission spectrum, a shift in the optical resonance of 15 nm was achieved.

## 2. Materials and Methods

All chemicals were obtained from suppliers and used without further purification. Ammonium hydroxide (NH_4_OH, 30%), hydroxylamine hydrochloride (NH_2_OH·HCl, 98%), polyacrylic acid (PAA, Mw: ~100k, 35 wt. % in H_2_O), and sodium borohydride (NaBH_4_, 98%) were purchased from Sigma-Aldrich (St. Louis, MO, USA). Dimethylformamide (DMF, 99.5%) and hydrogen tetrachloroaurate(III) trihydrate (HAuCl_4_·3H_2_O) were purchased from Alfa Aesar (Ward Hill, MA, USA). Fluorine doped tin oxide (FTO) substrates were purchased from STAREK Scientific Co., Ltd. (Taipei, Taiwan (R.O.C.)). To fabricate Au NIFs, the FTO substrates with a size of 3 × 3 cm were first rinsed with pure water, and then ultrasonically cleaned in acetone, methanol, and pure water for 10 min in each solvent. Then, a carboxylic acid-rich polymer of PAA was pre-coated onto the substrates to improve the adhesion of Au seeds to the FTO substrates. Typically, 200 μL of DMF solution containing PAA (0.7% *w*/*w*) was dropped onto a FTO substrate, followed by spin coating at 1500 rpm for 20 s. The speed was subsequently increased to 2500 rpm for 180 s. Afterwards, the FTO substrate with the PAA coating was annealed in an air atmosphere at 180 °C for 1 h. To generate the Au seed layer, the FTO substrate with the PAA coating was immersed in a 45 mL mixture containing HAuCl_4_ (3 mM) and NH_4_OH (0.3 M) at 10 °C for 20 min to deposit Au^3+^ ions. After rinsing with pure water, the FTO substrate was immersed in a NaBH_4_ (1 mM) solution at 10 °C for 2 min to reduce the adsorbed Au^3+^ ions to Au seeds. To promote the growth of the Au NIF, the FTO substrate with the Au seed layer was immersed in 50 mL of a growth solution containing HAuCl_4_ (0.5 mM) and NH_2_OH (0.5 mM) at 30 °C for 5 min. After washing with pure water, the FTO substrate with the Au NIF was subsequently blow-dried with nitrogen and stored at room temperature in a vacuum and dark environment for further measurements. SEM images of the Au NIF were acquired on a field emission SEM (S-4800, Hitachi Ltd., Tokyo, Japan) with an acceleration voltage of 10 kV. An arbitrary function generator (Tektronix AFG-3022B, Beaverton, OR, USA) was also employed to apply various electrical frequency-modulated voltages. In addition, transmission spectra of the LC/Au NIFs were obtained with a fiberoptic spectrometer (Ocean Optics HR 2000^+^, Ocean Optics, Largo, FL, USA). All experimental data were acquired at 26 ± 1 °C.

## 3. Results and Discussion

Figure 1a,b shows scanning electron microscopic (SEM) images and the corresponding magnified region of Au NIFs fabricated on a FTO coated glass substrate. The Au NIF was fabricated with a facile seed-mediated growth approach in the solution phase. Au NIFs with elongated nanostructures and winding inter-island gaps were characterized by SEM, and the average nanoisland size and gap distance were calculated from SEM images using ImageJ software. Built-in functions of particle analysis and watershed segmentation were used to determine the average island size and gap distance (Figure 1c,d). The Au NIF revealed an average nanoisland size of 6.6 × 10^4^ ± 1.8 × 10^4^ nm^2^ and an average inter-island gap distance of 69.3 ± 32.1 nm.

The substrate with the nanostructured surface served as the bottom substrate for the LC cell as schematically shown in Figure 2. The other substrate was a typical FTO-coated glass slide spin-coated with a PI alignment layer and treated with mechanical buffing. The assembled cell had a thickness of ~10 μm, as determined by silica spacers. E7 LC material was introduced into the empty cell. The top and bottom electrodes permit AC voltage to be applied across the cell thickness. Nonpolarized white light was successively transmitted through the top glass substrate and LC layer, and then was coupled to the Au NIF with plasmonic resonance. The LSPR can be adjusted by the dielectric constant of the surrounding LC material which is determined by the reorientation of the LC director by the externally applied electric field. Figure 3 shows optical textures of the tunable LC/Au NIF device with various applied voltages under crossed polarizers. We observed that the optical textures of the LC/Au NIFs were in a planar arrangement at 0 V. The LC/Au NIF textures were dark and bright when the angle between the rubbing direction and the polarizers was 0 and 45, respectively. When we increased the applied voltage, the direction of the LC molecules tended to reorient toward the direction of the electric field, and the colors of the optical textures gradually changed. The brightness slowly dimmed because of the decreasing birefringence.

In addition, the experimentally measured nonpolarized absorption spectrum of the LC/Au NIF as a function of applied voltages is shown in Figure 4. In Figure 4a, one can observe a relationship between the applied voltage and resonant wavelength in an AC field. A greater voltage was required to reorient the LC molecules near the nanostructure due to the strong anchoring forces of the nanostructure. That is why the tuning range of the active plasmonic device was limited in the past. However, we used the Au NIF which has a spherical bulge. With the random spherical shape of the Au NIF, the LC molecules reduced the arrangement of the groove effect. Consequently, the direction of the LC molecules could more easily be tilted up by applying an electrical field. The interface effect between gold and LC molecules is studied in the literature [17]. In Figure 4a, when the incident light went through the planar LC layer at 0 V, the effective refractive index near the Au NIF experienced by the nonpolarized incident light was around (*n_e_* + *n_o_*)/2, and the resonant wavelength was 477 nm. The LC molecule tilted under applied voltages of 2, 4, 6, and 8 V, and the resonant wavelength was red shifted. When the applied voltage was 8 V, the LC molecules almost completely reoriented themselves in a direction normal to the substrate surface, and the effective refractive index near the Au NIF was close to the ordinary refractive index, n_o_. Simulation results in previous research also showed that the resonant wavelength increased with a decreasing reflective index [18]. The curvature of LSPR to refractive index changes by using Mie’s theory, as well as the analytical expression of the LSPR curvature to the refractive index is proposed theoretically in the literature [19]. However, the resonant wavelength did not change when the applied voltage exceeded 8 V, because a quasi-homotropic LC was exhibited when the voltage exceeded 8 V. Moreover, the absorbance of the Au NIF decreased with increasing electrical voltage, because of a reduction in the refractive index. Spectra of the LC/Au NIFs with applied DC voltages of amplitudes 0~8 V are shown in Figure 4b. Both the wavelength and absorbance tunability of the LC/Au NIFs were not obvious under DC operation, because DC voltage possesses an ionic effect, which makes LC molecules insensitive to the applied voltage [20]. In the past, the birefringence change of LC induced resonance shifting is not significant. However, the spherical Au structure makes LCs easier to change arrangement nearing the interface and to shift the resonance.

Maxwell’s equation is solvable for spherical particles, as first proposed by Gustav Mie in 1908, and further developed by Bohren and Huffman. Thus, the absorption and scattering of spherical particles can be calculated from Mie’s theory [21]. The wavelength- and refractive index-dependent absorption A (λ, n) of the LC/Au NIF can be derived from Mie’s theory:$$A(\lambda ,n)=\frac{18\pi Vn_{eff}^{2}Nl}{\lambda }\frac{\left(a_{2}+b_{2}\lambda \right)}{{\left(a_{2}+b_{2}\lambda \right)}^{2}\left(b_{1}+a_{1}+2n_{eff}{}^{2}\right)+1}$$
where *a*_1_, *a*_2_, *b*_1_, *b*_2_, and *π* are constants, *V* is the volume of the particle, *N* is the density of electrons, *λ* is the wavelength of light, *l* is the length of the optical path, and *n*_eff_ is the effective refractive index of the surrounding medium. In addition, the effective refractive index (*n*_eff_) of the surrounding medium, which consisted of LC molecules in this case, can be described by:$$n_{eff}(\theta )=n_{e}\sin^{2}\theta +n_{o}\cos^{2}\theta$$
where *θ* is the angle between the direction of the LCs and the direction of the incident light. Based on the equation, the effective refractive index is sensitive to the tilting angle of the LCs. According to these equations, the absorption of the LC/Au NIF decreases with a reduction in the effective refractive index (or increasing applied voltage), which is the same behavior as in Figure 4a,b. Figure 5a,b shows the absorbance and wavelength tuning of the LC/Au NIFs with various applied voltages of amplitudes 0~8 V_rms_. From Figure 5a, one can see that the absorbance was highest at 2 V_rms_, because the LC molecules are better aligned under a small applied voltage. The greatest portion of (*n_e_* + *n_o_*)/2 of the LCs surrounding the Au NIF is demonstrated. The value of (*n_e_* + *n_o_*)/2 gradually changed to n_o_, resulting in a decrease in the absorbance and a red shift of the LSPR wavelengths. Based on the equations, the tilting angles of the LCs (complementary to angle *θ*) had a great influence on the absorbance and wavelength of LC/Au NIFs. Dynamic measurements of the tilting angle, namely, the polar angle measured from the substrate plane, were carried out using a typical crystal-rotation method [22]. Figure 6 displays variations in the tilting angle of the LC/Au NIFs with applied voltages measured by the typical crystal-rotation method. The angle dramatically increased from ~15 to ~80° and was finally saturated at ~80°. Compared to the well-known LC plasmon resonance device, the most advantageous feature of our LC/Au NIF is the nonpolarized, tunable LSPR. We found that the wavelength tunability of ~15 nm and absorbance tunability of ~0.024 were shown based on the LC/Au NIFs.

## 4. Conclusions

In summary, optical properties of the Au NIFs covered with LC materials are introduced. We demonstrated that the resonance wavelength-dependent refractive index and orientation of the LCs in the hybrid LC/Au NIFs significantly changed when various voltages were applied. The hybrid LC/Au NIF structure enabled the active control of the resonance wavelength of the spectrum in the metallic nanoisland. A combination of the nanostructured surface and birefringent LC opens the door for new electro-optical devices, tunable devices, nanoscale sensors, and additional applications.

## Figures and Tables

**Figure 1 nanomaterials-10-00290-f001:**
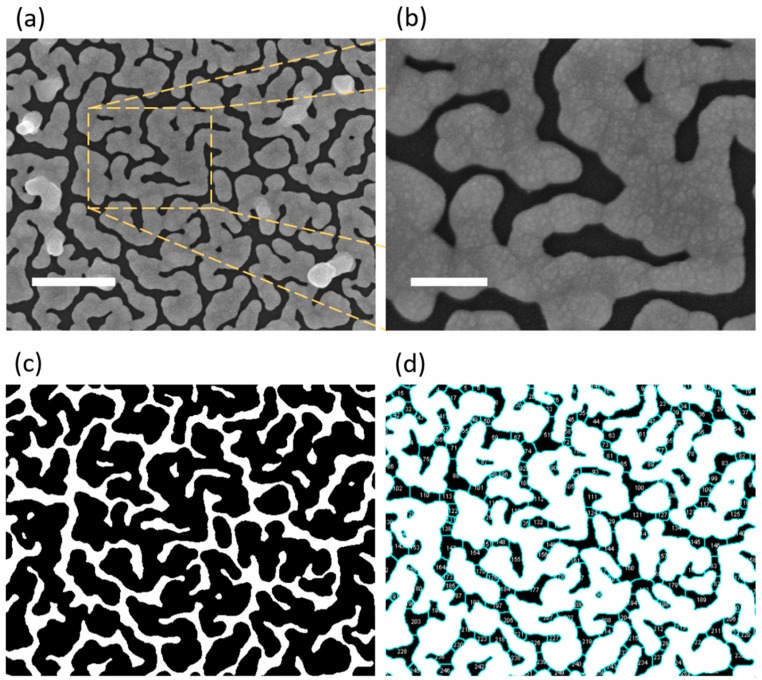
(**a**) SEM image of the gold nanoisland film (Au NIF) (scale bar = 500 nm) and (**b**) the selected magnified area (scale bar = 150 nm). (**c**) The converted binary image of the Au NIF used for the calculation of average nanoisland size from (**a**). (**d**) An inverted image of (**c**) was applied to determine the average inter-island gap distance, where the blue line shows gaps after the watershed segmentation process.

**Figure 2 nanomaterials-10-00290-f002:**
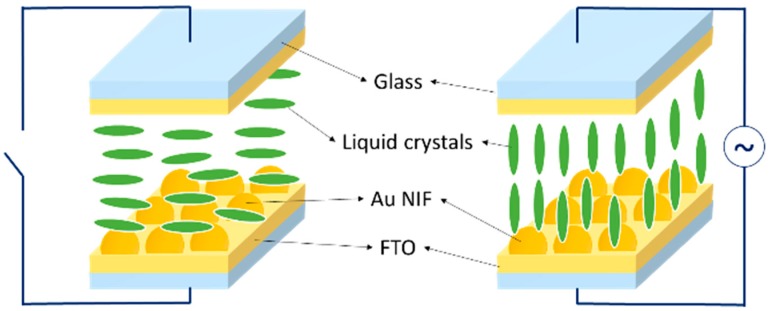
Schematic of the liquid crystal (LC)/gold nanoisland film (Au NIF) devices. An applied electric field across the cell causes the LC to reorient and changes the wavelengths of the resonant light.

**Figure 3 nanomaterials-10-00290-f003:**
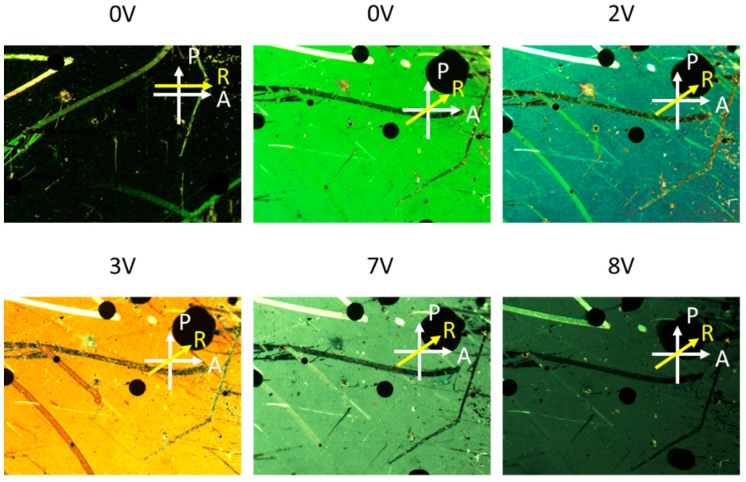
Optical textures of tunable liquid crystal (LC)/gold nanoisland film (Au NIF) with various applied voltages under crossed polarizers. White arrows indicate the transmission axes of the polarizer and analyzer, while the yellow arrow indicates the rubbing direction.

**Figure 4 nanomaterials-10-00290-f004:**
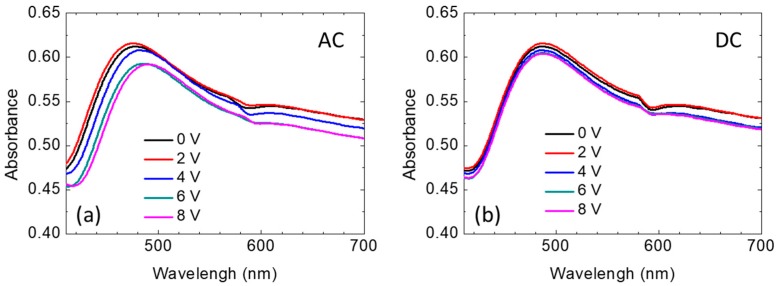
Nonpolarized spectra of the liquid crystal (LC)/gold nanoisland film (Au NIF) with applied voltages of amplitudes 0~8 V under (**a**) Alternating current (AC) and (**b**) Direct current (DC) voltage.

**Figure 5 nanomaterials-10-00290-f005:**
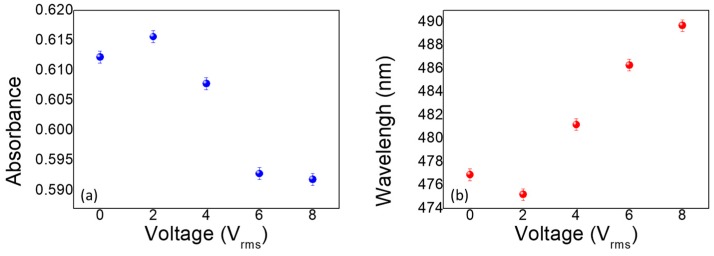
Absorbance and peak wavelength of the liquid crystal (LC)/gold nanoisland film (Au NIF) with various applied voltages of amplitudes 0~8 V_rms_.

**Figure 6 nanomaterials-10-00290-f006:**
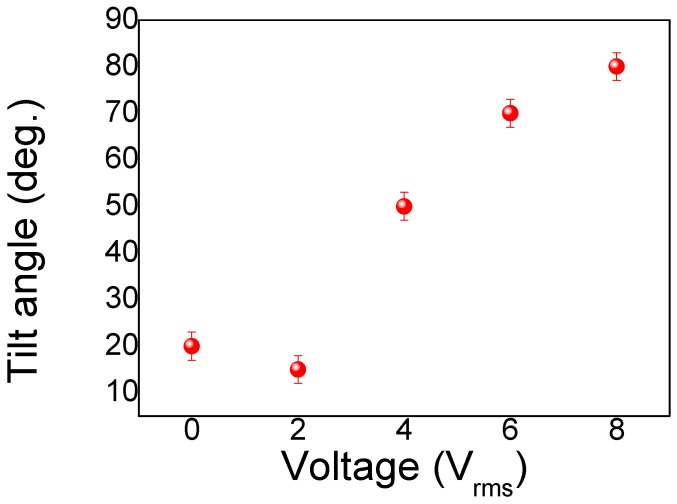
Variation of the tilting angle of the liquid crystal (LC)/gold nanoisland film (Au NIF) with various applied voltages.

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
