# Peer review of "Optical Properties of Electrically Active Gold Nanoisland Films Enabled with Interfaced Liquid Crystals"

_nanomaterials, 2020, doi:10.3390/nano10020290_

Round 1

Reviewer 1 Report

This paper reports on the realization of an electrically controlled plasmonic filter obtained by combining gold islands and nematic liquid crystal. The combination of plasmonic nanostructures and liquid crystals has been intensively studied in the past 10 years. For this reason, the novelty is very low. Moreover, it is very unclear the statement:” Simulation results in previous research also showed that the resonant wavelength increased with a decreasing reflective index [15].” According to the Mie theory there is a blue shift of the plasmonic resonance if the refractive index of the surrounding medium is decreased. Another important aspect is the definition of “nano-islands”.  Figure 1b shows very large domains (more than 100 nm). For this reason, they cannot be called “nano”. In addition, it is not clear what is going on at the interface between gold and LC molecules (since there is no alignment layer). Lastly, I suggest to read more carefully the literature on the subject. There are so many high level papers not even included in the reference list. For all the aforementioned reasons, I cannot recommend this paper for publication.

Reviewer 2 Report

In this manuscript, the authors demonstrated the development and optical properties of a gold nanoisland film covered with a liquid crystal material. Using both numerical and experimental investigations, it is shown that the optical characteristics of the tailored system can be tuned and modified by applying external bias. Although the results are validated correctly, there are some important points that must be addressed carefully. First of all, the writing style of the manuscript needs to be improved and some grammatical mistakes must be corrected and polished. In addition, In Figure 4, the plotted absorption spectra does not show strong dependency on the bias variations. It is claimed that by increasing the bias, the resonance was shifted, however, the shift is not significant. This must be discussed comprehensively.

Again in the same graph, I was expecting to observe the interband transitions for gold film starting from 450 nm rather than UV band. This must be explained.

Reviewer 3 Report

This paper reports optical properties of Au nanoisland film covered with liquid crystal.  This paper containes interesting results for many readers.  I think this paper acceptable without any changes.

Round 2

Reviewer 1 Report

This paper is not suitable for publication. 

Author Response

We thank the reviewer for good suggestion. The paper first show the active tuning of Au NIF metallic nanostructure embedded in the LC. We think it is important for  academic research.

Reviewer 2 Report

Acceptable. However, the axes in most of diagrams must be prepared well and the starting and ending values must be indicated.

Author Response

We thank the reviewer for such good suggestions.